# Gallbladder Neuroendocrine Neoplasms in Dogs and Humans

**DOI:** 10.3390/vetsci11080371

**Published:** 2024-08-12

**Authors:** Nadia Kelly, Yen-Tse Wu, Andrea N. Johnston

**Affiliations:** 1Department of Veterinary Clinical Sciences, School of Veterinary Medicine, Louisiana State University, Baton Rouge, LA 70803, USA; nkelly5@lsu.edu (N.K.); ywu95@ncsu.edu (Y.-T.W.); 2Emergency & Critical Care, College of Veterinary Medicine, North Carolina State University, Raleigh, NC 27607, USA

**Keywords:** neuroendocrine tumors, neuroendocrine carcinomas, canine, hepatobiliary cancer

## Abstract

**Simple Summary:**

Cancers of the gallbladder arise in dogs and humans. Neuroendocrine neoplasms are the rarest type of gallbladder cancer in both species. This review summarizes the characteristics of canine and human gallbladder neuroendocrine neoplasms with the objective of comparatively modeling clinical and diagnostic features of these spontaneously occurring neoplasms.

**Abstract:**

Gallbladder neuroendocrine neoplasms (GB NENs) are among the rarest cancers reported in humans and dogs. This review provides a detailed review of the canine GB NEN literature and an interspecies comparison of demographics, clinical pathophysiology, pathobiology, and therapeutic response of GB NENs. The aim of this work is to explore the relevance of dogs as a spontaneous model for human GB NENs.

## 1. Introduction

Historically referred to as carcinoids (from karzinoide, meaning “carcinoma-like”), neuroendocrine neoplasms (NENs) comprise a heterogeneous group of tumors derived from cells of the neuroendocrine system. Neuroendocrine (NE) cells, which may be gland-forming or diffusely distributed, characteristically express features of both endocrine cells and neurons [1]. Functionally, NE cells synthesize bioactive amines, peptides, and hormones. Similar to serotonergic neurons, bioactive molecules are stored within dense core granules that are released in response to receptor-mediated stimuli [2]. The specific hormonal products secreted by NE cells differ depending on their tissue niche, and NENs arising from anatomically distinct sites often exhibit diverse biologic behavior [1,3,4,5].

NENs account for approximately 0.5% of all human malignancies. The majority of these neoplasms occur in the gastrointestinal and respiratory tracts, a pattern of distribution that reflects the high concentration of dispersed neuroendocrine cells within these organs. Less common sites of origin include the pancreas, skin, urogenital tract, and hepatobiliary system [6]. Historically, NENs were classified according to their synthetic hormone profile, but this classification scheme was complicated by the complexity of NEN phenotypes. Nomenclature guidelines by the World Health Organization (WHO) have refined the terminology by dividing NENs into categories based on histopathologic features, including the well-differentiated neuroendocrine tumor (NET), poorly differentiated neuroendocrine carcinoma (NEC), and mixed neuroendocrine carcinomas which have features of NENs and non-neuroendocrine neoplasms (MiNEN) [7,8]. NETs are further subdivided into grades 1–3, with grades 1 and 2 exhibiting indolent growth and a favorable prognosis, while grade 3 NETs have a high Ki67 index (>20%) and carry a poor prognosis. NECs, which are poorly differentiated, are considered high-grade. MiNENs are graded according to both their neuroendocrine and non-neuroendocrine components [7,8].

Reports of NENs in the veterinary literature are uncommon, although incidence has increased since the turn of the century as advances in diagnostic imaging have enabled earlier diagnosis of these neoplasms. In dogs, NENs of the pancreas, paraganglia, pituitary gland, and thyroid gland predominate, while less common locations include the gastrointestinal tract, the bronchopulmonary tree, the skin, and the hepatobiliary system [9,10,11,12,13,14,15,16,17,18].

## 2. Characteristics of Neuroendocrine Cells

The neuroendocrine system comprises a wide range of cells found throughout most mucosal organs in the body. NE cells are present in large aggregates comprising solid organs such as the adrenal medulla and pituitary gland, small focal aggregates such as those of the pancreatic islets or pulmonary neuroepithelial bodies (NEBs), and individualized cells scattered throughout the mucosa of numerous tissues. While the majority of the diffuse neuroendocrine cells of the body localize to the gastrointestinal tract, pancreas, and respiratory system, lower numbers of these cells can be found throughout many other tissues, including the urogenital system, the biliary tract, the skin, and the thymus. While NE cells do exhibit phenotypic differences depending on their specific tissue localization, they generally share a number of common characteristics, including the production and secretion of hormones and neurotransmitters. In response to environmental stimuli or neural input, neuroendocrine cells are stimulated to secrete granules containing bioactive molecules that may act on distant tissue sites (endocrine), neighboring cells (paracrine), or the cell of origin (autocrine) [19].

## 3. Neuroendocrine Neoplasms: Nomenclature and Grading

In 1963, the first formal classification scheme for NENs was established using the embryologic origin of the primary organ—specifically the foregut (respiratory tract, stomach, proximal duodenum, pancreas, hepatobiliary system), midgut (duodenum, ileum, appendix, ascending colon), or hindgut (transverse and descending colon, rectum). However, the wide variation in the behavior of NENs even within these embryonic segments limited the clinical utility of this classification system. Additionally, the notion that carcinoids of each embryonic segment had a distinct histologic appearance was later discredited, and it is now known that considerable histopathologic overlap exists between all NEN subtypes [8].

In 1999, the World Health Organization (WHO) attempted to clarify the nomenclature of NENs by establishing a classification scheme for pulmonary NENs consisting of four categories. Typical (low-grade) carcinoids were well-differentiated NENs consistent with the prototypic carcinoid first described by Oberndorfer, while atypical (intermediate-grade) carcinoids had a slightly more aggressive clinical course while still appearing overall well differentiated. High-grade NENs were separated into the categories of large-cell neuroendocrine carcinoma (LCNEN) and small-cell lung carcinoma (SCLC), both of which appeared poorly differentiated and exhibited malignant characteristics [2]. The following year, the term “carcinoid” was dropped from gastrointestinal and pancreatic NENs altogether in favor of the broad category of “neuroendocrine neoplasm”. Typical and atypical carcinoids were reclassified as grade 1 (G1) or grade 2 (G2) neuroendocrine tumors (NETs), respectively, while the LCNEC and SCLC were now considered subtypes of neuroendocrine carcinomas (NECs). In 2010, the NET/NEC classification system was expanded to encompass all neoplasms of neuroendocrine origin. Criteria for grading were also established for gastrointestinal and pancreatic NENs (GEP-NENs) using mitotic and Ki67 indices [20].

Over the following decade, evidence emerged for another category of NEN that did not fit within the 2010 WHO guidelines. Morphologically well-differentiated NENs with a Ki67 index exceeding 20% presented a diagnostic conundrum; while the histopathologic features would qualify these tumors as NETs, their proliferation index and clinical course were consistent with an NEC. The pancreas was the most common site for these “high-grade NETs”, and the 2017 WHO guidelines for endocrine neoplasms expanded the grading system for pancreatic NENs to include a NET G3 category. In 2019, the use of the G3 category was extended to all GEP-NETs [21].

Specific guidelines exist for grading NETs in each anatomic niche, but grading is generally based on mitotic index, Ki67 index, and/or the presence of necrosis. NECs are inherently considered to be high-grade. It was once accepted that NETs arose from differentiated neuroendocrine cells and that NECs were the result of NET progression to a malignant phenotype (sometimes referred to as the “NET-NEC sequence”); however, cell lineage tracing and genetic analyses have discredited the NET-NEC theory. The current understanding is that NENs are the result of neoplastic transformation of a stem cell precursor, and the resultant type of NEN depends on the stage of differentiation of that precursor (i.e., NECs arise from pluripotent stem cells, while NETs arise from committed neuroendocrine stem cells). No strong evidence exists for the progression of a NET to a NEC, and they likely represent separate and distinct entities [22,23].

## 4. Epidemiology and Anatomic Distribution of Neuroendocrine Neoplasms

Though once considered to be rare, the reported incidence of NENs in humans has been steadily increasing. The rising number of cases is likely multifactorial, owing to improved awareness of these tumors and more accurate classification with modern immunohistochemical techniques. According to data from the National Cancer Institute’s Surveillance, Epidemiology, and End Results (SEER) Program, the incidence rate of NENs in the United States increased 6.4-fold from 1973 (1.09 per 100,000) to 2012 (6.98 per 100,000), with increases distributed across all anatomic sites [24]. The greatest increases were observed for gastric (15-fold) and rectal (9-fold) NENs, which have been attributed to the widespread modern use of endoscopic procedures [24,25,26].

The gastrointestinal tract is the most common site of NEN development, accounting for 67.5% of NENs in one retrospective analysis of 13,715 cases. Of gastrointestinal NENs within that study, the small intestine was the most common site (41.8%), followed by the rectum (27.4%) and stomach (8.7%). The respiratory tract was found to be the second most common organ system of NEN origin (25.3%), with the remainder of cases distributed among less common locations, including the pancreas, urogenital tract, thymus, and hepatobiliary system. A slight overall female predominance was found for NENs as a whole (55.1%), with the highest female-to-male ratios observed in the stomach, colon, appendix, bronchopulmonary system, and gallbladder. Male predominance was noted in esophageal and thymic NENs. Metastatic disease was present in 12.9% of patients at the time of diagnosis, and the overall 5-year survival rate was 67.2%, although the range in overall survival was broad. Five-year survival rates were best for rectal (88.3%), bronchopulmonary (73.5%), and appendiceal (71.0%) NENs, while the poorest prognosis was observed with pancreatic (37.5%) and hepatic (18.4%) tumors [27,28].

The epidemiologic characteristics of NENs in dogs have not been well characterized, but common sites include the endocrine pancreas, the thyroid gland, the adrenal medulla and paraganglia, and the pituitary gland [29,30,31,32,33]. Canine thyroid tumors have a prevalence of 1–4% [34]. In dogs, insulinomas comprise ~1–2% of all pancreatic neoplasms [35,36]. Pheochromocytomas and paragangliomas account for 0.01 to 0.1% and 0.2%, respectively, of canine tumors [32]. Pituitary tumors account for 13% of intracranial tumors in dogs [33]. Less commonly, NENs have been documented in the hepatobiliary, gastrointestinal, and respiratory tracts [37,38]. No definitive risk factors have been identified.

## 5. Gallbladder Neuroendocrine Neoplasms

The mechanisms and regulation of neuroendocrine differentiation in the hepatobiliary tract have not been extensively clarified. It is now accepted that these sparse NE cells are of endodermal origin, derived from local populations of multipotent stem cells, rather than arising from the neural crest as was originally postulated. However, the exact molecular pathways governing their differentiation and the role these cells play in pathologic states is an area of ongoing study.

The origin of gallbladder NENs has been an issue of debate, as there are no neuroendocrine cells within the normal gallbladder mucosa. The theories proposed are that GB NENs arise either from NE cells in the gallbladder neck or from gastric or intestinal metaplasia, which may occur in the gallbladder secondary to conditions such as cholelithiasis. The fact that GB NENs are frequently observed in human patients with a history of cholelithiasis has been viewed as supporting evidence for the latter theory; while cholelithiasis is uncommon in dogs, 17.1% of the dogs described herein had choleliths concurrent with gallbladder NEN [10,39]. Yet given the more recent revelations regarding the multipotent stem cell origin of most NENs, GB NENs may be derived from local progenitor cells rather than differentiated cells [10,23,39,40,41,42].

### 5.1. Diagnosis of Gallbladder NENs: Cytopathology

Neuroendocrine neoplasms across anatomic locations share similar cytologic features in humans and dogs (Figure 1). Samples are commonly composed of loosely cohesive clusters of round to polygonal cells with indistinct borders, often forming palisades or rosettes. Characteristically, these samples tend to contain large numbers of intact free nuclei dispersed throughout a diffuse background of pale basophilic cytoplasm. In the majority of cases, anisokaryosis and nuclear atypia are minimal. Nuclear-to-cytoplasmic ratios are variable, and the cytoplasm may be lightly discretely vacuolated. Less commonly, nuclear pleomorphism may be more pronounced; karyomegaly and nuclear pseudoinclusions are rarely observed [5,8,43]. For most cases, specific diagnosis requires interpretation of cytopathologic findings in conjunction with anatomic location and clinical features.

### 5.2. Diagnosis of Gallbladder NENs: Histopathology

In general, the histopathologic characteristics of NENs in both humans and dogs are relatively consistent across anatomic niches. These neoplasms are composed of round to cuboidal cells arranged in one of several characteristic patterns, which include nests, rosettes, and solid cords interspersed by fibrovascular stroma. Well-differentiated NENs are composed of uniform polygonal cells, which can be arranged in solid nesting architecture, trabecular or ribbon-like morphology, or glandular patterns. Combinations of these patterns are sometimes observed, and palisading may be present near the periphery of nested arrangements. Abundant fibrous stroma is present in some cases, but more commonly, tumor cells are interspersed by a fine fibrovascular stroma [5,8,43].

Poorly differentiated NECs may be of large- or small-cell phenotype. Large-cell NECs (LCNECs) have low nuclear-to-cytoplasmic ratios, well-defined cell borders, and clear to eosinophilic cytoplasm that may be granular. In contrast, small-cell NECs (SCNECs) have indistinct cell borders and small, hyperchromatic nuclei with scant cytoplasm. In both subtypes, cells are arranged predominantly in solid sheets or nodules, and necrosis is common [8].

### 5.3. Diagnosis of Gallbladder NENs: Immunohistochemistry

A key characteristic enabling the identification of neuroendocrine cells is the presence of a distinct profile of cell markers. Broad-spectrum markers that can be used to identify neuroendocrine differentiation include chromogranin A (humans and dogs), synaptophysin (humans and dogs), and insulinoma-associated protein 1 (only validated in humans) [8,10,44].

Granins are acidic hydrophilic glycoproteins found in large dense-core vesicles (LDCVs) of neuroendocrine cells [45]. They are initially synthesized at the rough endoplasmic reticulum (RER), before being transported to the Golgi complex, where they aggregate at the trans-Golgi network membrane and induce budding of LDCV. Granins also serve as precursors for numerous biologically active peptides, which induce a wide range of physiologic effects. The major granins include chromogranin A (CgA), chromogranin B, and secretogranin II. CgA is the most abundantly expressed granin, comprising 50% of the soluble protein content of adrenal medullary chromaffin secretory granules [1,46]. CgA is very broadly expressed in neuroendocrine cells and is considered the most specific of the broad-spectrum neuroendocrine markers. However, since granins are found almost exclusively within secretory granules, poorly differentiated neuroendocrine cells with few mature granules may be negative for this marker. In some cases, a fine paranuclear dot-like pattern of positivity may be observed.

Synaptophysin (SYN), a transmembrane glycoprotein found on smaller synaptic-like microvesicles (SLMVs), is expressed by both normal and neoplastic NE cells [8,45]. Its role in SLMV trafficking is still incompletely understood, but it appears to play a role in vesicle recycling following exocytosis. SYN is considered the most sensitive marker for both well-differentiated and poorly differentiated NENs but lacks complete specificity, as other neoplasms (e.g., adrenocortical carcinomas, neuroblastomas) may also be positive [45].

Insulinoma-associated protein 1 (INSM1), a zinc-finger transcription factor, is a relatively new broad-spectrum NE marker. Originally isolated from pancreatic insulinoma cells, INSM1 acts to repress transcriptional activity, regulate cell cycle entry, and control the expression of neuroendocrine phenotype [47]. INSM1 plays a key role in the development of NE cells, and studies have demonstrated that it matches both the sensitivity of synaptophysin and the specificity of chromogranin A, making it a promising marker for the diagnosis of NENs. Additionally, the nuclear staining pattern of INSM1 is more readily identified in comparison to the cytoplasmic expression of CgA and SYN [48]. To date, large-scale evaluation of the utility of INSM1 as an NE marker in dogs has not been reported.

Other broad-spectrum NE markers that have fallen out of common use include neuron-specific enolase (NSE), protein gene product 9.5 (PGP 9.5), and CD56 [3,43,49]. NSE, an isozyme of the glycolytic enzyme enolase, is found in the cytoplasm of normal and neoplastic NE cells [50]. NSE has classically been used as a marker of neuroendocrine differentiation, but its clinical utility is hampered by overall poor specificity, as anti-NSE antibodies have been shown to cross-react with enolase expressed in non-NE tissue such as striated muscle [51]. Additionally, NSE positivity has been detected in some non-NE neoplasms of the pancreas. PGP 9.5 is a protease involved in the processing of ubiquitin precursors and is expressed by neurons and neuroendocrine cells; however, PGP 9.5 expression has also been documented in pancreatic adenocarcinomas and plasma cell neoplasms. CD56, also called neural cell adhesion molecule (N-CAM), is found in a wide range of cells including T cells, natural killer cells, thyrocytes, adrenocortical cells, and NE cells. It has a high sensitivity for neuroendocrine differentiation, but its specificity is poor, and positivity may be observed in numerous non-NE neoplasms [49]. CD56 is still utilized sporadically in conjunction with CgA and SYN for the diagnosis of pulmonary NECs, but NSE and PGP 9.5 are not currently recommended for diagnostic use [8,52].

## 6. Gallbladder NENs in Humans

First described in 1929, NENs of the human gallbladder are rare, comprising only 0.5% of all NENs and approximately 2% of gallbladder neoplasms. They have a distinct female predominance, with a female-to-male ratio of 4:1, and while the low number of documented cases and lack of consistent nomenclature in past studies limits the interpretation of their biologic behavior, they are generally considered to be aggressive neoplasms with a poor prognosis. Retrospective analyses have reported that NECs comprise 69.9–90.2% of GB NENs, with the majority being small cell carcinomas (SCCs). Abdominal pain is the most common presenting complaint, occurring in approximately half of patients, while less consistent clinical signs include vomiting, icterus, and abdominal distension [42,53].

Many human patients with GB NENs are initially asymptomatic or have non-specific clinical signs, making early identification of these neoplasms challenging; metastases are reported in 54–68% of cases at the time of diagnosis. Clinicopathologic features consistently correlated with prognosis include tumor grade, the presence of metastatic disease, and advanced age. Tumor size has also been reported as a prognostic feature, with more than 70% of tumors > 2 cm presenting with local or distant metastases [54]. Prognostic information in human GB NENs is conflicting, complicated by lack of stratification by grade and the use of inconsistent terminology prior to the 2010 WHO reclassification of NENs; however, the prognosis in most cases is poor, with reported median survival times of 3–24 months. Grade 1 and G2 NETs have a more favorable prognosis if surgical excision is performed, but specific survival data for these grades are scarce [53].

## 7. Gallbladder NENs in Dogs

Primary gallbladder NENs in dogs were first described in 1988, and sporadic case reports or case series of gallbladder NENs have appeared in the literature since that time [10,11,18,39,44,55,56,57,58,59,60,61]. Grossly, these masses are firm, solitary, and variable in size. Necrosis and hemorrhage are common histopathologic features, and concurrent gallbladder pathology is frequently present, including cholecystitis, cystic mucosal hyperplasia, and cholelithiasis. Notably, the findings of these studies indicate that primary gallbladder NENs may carry a more favorable prognosis compared to gallbladder NENs in humans; however, the biologic behavior of canine NENs is poorly understood due to the lack of histologic grading [10].

### 7.1. Canine Patient Demographics

A search for the publications describing canine GB NENs was conducted through the National Center for Biotechnology Information PubMed database using the search terms: neuroendocrine tumor, neuroendocrine carcinoma, neuroendocrine cancer, OR carcinoid, AND gall bladder, gallbladder, biliary, OR hepatobiliary, AND canine, dog, OR veterinary. Thirty-five dogs with histologically confirmed gallbladder neuroendocrine neoplasms were identified in 13 publications [10,11,18,39,44,55,56,57,58,59,60]. Demographic data, clinical signs, histologic findings, and survival were summarized when available (Table 1). The median age at diagnosis was 9.0 years (yr, range: 5.0–13.0 yr). Sex distribution identified a male predominance with 26 (74.29%) males including 10 (28.57%) neutered males, 1 (2.86%) intact male, and 15 (42.86%) dogs with unknown castration status. Only 9 (25.71%) female dogs were identified, including 5 (14.29%) spayed females and 4 (11.43%) females with unknown ovariohysterectomy status. Twenty breeds were identified. The most common breeds were Boston terriers (25.71%), mixed breed dogs (11.43%), French bulldogs (8.57%), bull mastiffs (5.71%), and bichon frises (5.71%). One dog (2.86%) was identified from the following breeds: golden retriever, Irish wolfhound, American Eskimo, bulldog, English bulldog, Doberman pinscher, poodle, Cocker spaniel, shih tzu, Keeshond mix, beagle mix, Rhodesian ridgeback, boxer, Australian shepherd, and Belgian Malinois.

### 7.2. Presenting Complaints and Physical Exam Findings

Presenting complaints were described in 34 dogs. The most common presenting complaint was vomiting (21 dogs, 61.76%), followed by vomiting with hematemesis (11 dogs, 32.35%), melena (6 dogs, 17.6%), anorexia (4 dogs, 11.76%), diarrhea (4 dogs, 11.76%), unspecified gastrointestinal signs (5 dogs, 14.71%), abdominal pain (3 dogs, 8.82%), fever (2 dogs, 5.88%), weight loss (2 dogs, 5.88%), lethargy (2 dogs, 5.88%), abdominal distention (2 dogs, 5.88%), hematuria (1 dog, 2.94%), respiratory signs (1 dog, 2.94%), and neurologic signs (1 dog, 2.94%). Five dogs (14.71%) were asymptomatic. Physical exam findings were reported in 13 dogs.

### 7.3. Hematological and Biochemical Profiles

The most common biochemical abnormalities included increased activities of ALT (22/25 dogs, 88%), ALKP (19/25 dogs, 76%), GGT (9/18 dogs, 50%), AST (8/20 dogs, 40%), and hyperbilirubinemia (4/24, 16.67%). The most common abnormalities reported on complete blood count were anemia in 20% (5/25) and leukocytosis in 12% (3/25) of dogs. In dogs that had leukocytosis, two were characterized by mature neutrophilia, and one was characterized by neutrophilia with left shift and monocytosis.

### 7.4. Grading and Mitotic Index

Histopathologic grading, based on the criteria described in Section 5.2 [59] and the WHO guidelines [44], was reported for 3 patients (Table 1), including 1 described as a low-grade neuroendocrine carcinoma, and 2 NET-grade 1 [21,43,44]. The mitotic index was reported for 22 patients (Table 1) [10,11,39,55,56,58]. Fifteen (68.18%) patients had measurements utilizing the unit “per 10 high power fields (HPF)”: 2/15 (13.33%) having <2 mitoses per 10 HPF and 13/15 (86.67%) having 2–20 mitoses per 10 HPF. In 4 dogs, the mitotic count was reported per 1 HPF: <1, 1, 1 to 2, and 1 to 3 mitoses per HPF. Three patients had unspecified numbers of mitoses (1 “scattered”, 2 “high”).

### 7.5. Immunohistochemical Profile

Immunohistochemical findings were available for a total of 22 dogs and are summarized (Table 2). Synaptophysin immunoreactivity was evaluated in 17 cases, all of which were positive (100%). Positive labeling for chromogranin A was found in 21/22 neoplasms (95.45%), while 18/19 (94.74%) were positive for neuron-specific enolase (NSE), and 13/15 (86.67%) were positive for gastrin. Immunostaining for PGP 9.5 was performed in two cases, and both were positive (100%) [55]. Negative results were found in all cases labeled for the epithelial marker, cytokeratin 7 (CK-7), (0/5, 0.0%) and the Schwann cell and melanocyte marker, S100 (0/2, 0.0%) [44,56,58]. Antigen Kiel 67 (Ki-67) proliferation index was quantified in 2 dogs, and the results were 0.04%, and 0.05% [44].

### 7.6. Canine Survival Data

Censored survival data were available for 26 dogs (Table 1). Fifteen patients (57.69%) were still alive at the termination of data collection. Of the 11 patients that died, 5 deaths were directly due to the GB-NEN, whereas 6 patients were euthanized due to various causes (1 intermittent vomiting and diarrhea post-cholecystectomy, 1 lymphoma, 1 neurologic disorder, 1 pulmonary masses, and 1 cardiac mass, and one dog whose cause of death was not reported). Overall median survival time (MST) was 730 days (range: 240–2191). Metastasis was reported in 6 of the 35 dogs (17.14%), with sites including the liver (*n* = 4), lungs (*n* = 2), and mesentery (*n* = 1).

### 7.7. Canine Therapy

Treatment information was available for 26 patients. Cholecystectomy was performed on 25/26 of dogs (96.15%). Other surgical procedures performed in addition to cholecystectomy included duodenotomy and common bile duct stenting in 2/26 (7.69%), and splenectomy in 2/26 (7.69%) of dogs [57,58,62]. In the two dogs that underwent splenectomy, histopathology revealed nodular hyperplasia and hemangiosarcoma (unknown if primary or metastatic), respectively [58]. One (1/26, 3.85%) patient underwent choledochotomy; one (1/26, 3.85%) patient underwent caudal liver lobectomy (nodular hyperplasia with acute hepatitis) [55,57]. One (1/26, 3.85%) patient underwent only a choledochotomy without a cholecystectomy [62]. Two (2/26, 7.69%) patients received chemotherapy (1 with 5 doses of carboplatin, 1 with doxorubicin for splenic hemangiosarcoma) following the cholecystectomy [44,58].

### 7.8. Summary Canine GB NENs

While the rarity of this neoplasm limits understanding of its behavior, the prognosis for canine patients with GB NENs appears to be good. In cases where mitotic index or Ki-67 has been reported, the majority of canine NENs can be classified as NETs. Mean survival times exceeding three years have been reported following cholecystectomy, which is consistent with reports of survival in human patients with completely excised G1 NETs.

It has been suggested that the production of gastrin by a portion of these neoplasms in dogs and the resultant clinical signs of vomiting and hematemesis may lead to an earlier diagnosis and contribute to a favorable prognosis; however, gastrointestinal signs have also been documented in a dog with a GB NEN that was negative for gastrin, indicating that the clinical presentation associated with this neoplasm is likely multifactorial [10,44].

## 8. Conclusions

NENs are a diverse group of neoplasms exhibiting significant behavioral heterogeneity, and the prognosis of primary gallbladder NENs in canine patients is better than that reported in humans. In contrast to NENs of the human gallbladder, which consist predominantly of NECs, the described histopathologic features and slower clinical course suggest that the majority of NENs in dogs are NETs. Improved histologic characterization of NENs in dogs, including more consistent reporting of proliferative indices, is needed to appropriately classify these tumors in the future. Further analysis of the molecular signatures of canine and human gallbladder NENs will contribute to improved understanding of their pathophysiology and prognosis. Furthermore, given the rarity of hepatobiliary NENs, a better understanding of these neoplasms will inform the development of therapeutic targets that may benefit both species.

## Figures and Tables

**Figure 1 vetsci-11-00371-f001:**
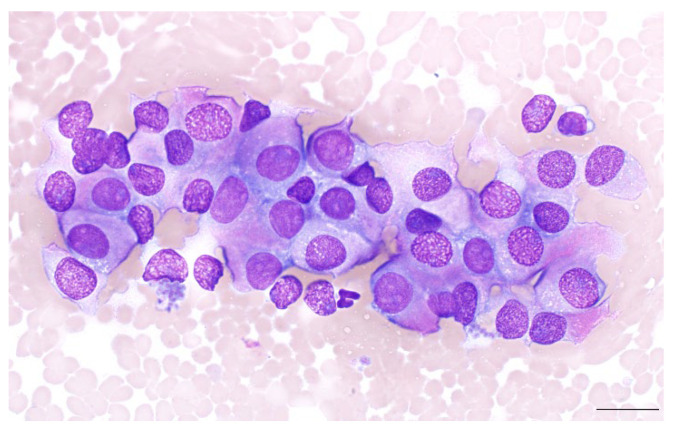
Gallbladder neuroendocrine neoplasm cytology. Photomicrograph of a fine needle aspirate smear from a gallbladder NEN in a dog. Polygonal epithelial cells demonstrate overall mild anisokaryosis and variably distinct cell borders. Wright–Giemsa stain; Scale bar = 20 µm.

**Table 1 vetsci-11-00371-t001:** Summary of patient demographics, grading and mitotic index, and survival data.

Dog	Breed	Age (Years)	Sex	Grade	Mitotic Index	Survival (Days)
1 [18]	Bull Mastiff	9	M	—	1–2 per 400× field	300+
2 [39]	Mixed breed	10	FS	—	<1 per 400× field	240
3 [11]	Mixed breed	10	M	—	High (number unspecified)	120+
4 [55]	Rhodesian Ridgeback	10	MC	—	1–3 per 1 HPF	5+
5 [55]	Beagle mix	12	FS	—	Scattered (number unspecified)	3+
6 [56]	Keeshond Cross	13	FS	—	<1 per 10 HPFs	365+
7 [57]	Bull Mastiff	8.5	FS	—	—	930
8 [58]	Shih Tzu	10	F	—	—	540+
9 [58]	Cocker Spaniel	10	M	—	1 per 1 HPF	300+
10 [59]	Irish Wolfhound	7.4	MC	Low-grade neuroendocrine carcinoma (carcinoid)	—	—
11 [62]	Boston Terrier	10	M	—	High (number unspecified)	1350
12 [44]	Boston Terrier	5	MC	NET G1	—	730+
13 [44]	Doberman Pinscher	7	MC	NET G1	—	730+
14 [60]	French Bulldog	*	*	—	—	—
15 [60]	French Bulldog	*	*	—	—	—
16 [60]	Boxer	*	*	—	—	—
17 [60]	American Eskimo	*	*	—	—	—
18 [60]	Boston Terrier	*	*	—	—	—
19 [60]	Golden Retriever	*	*	—	—	—
20 [60]	Bulldog	*	*	—	—	—
21 [60]	Australian Shepherd	*	*	—	—	—
22 [10]	*	*	*	—	10 per 10 HPFs	1926
23 [10]	*	*	*	—	6 per 10 HPFs	2191
24 [10]	*	*	*	—	3 per 10 HPFs	1021
25 [10]	*	*	*	—	4 per 10 HPFs	432
26 [10]	*	*	*	—	9 per 10 HPFs	1537
27 [10]	*	*	*	—	4 per 10 HPFs	1785
28 [10]	*	*	*	—	8 per 10 HPFs	1174
29 [10]	*	*	*	—	2 per 10 HPFs	1178
30 [10]	*	*	*	—	12 per 10 HPFs	1834
31 [10]	*	*	*	—	13 per 10 HPFs	1151
32 [10]	*	*	*	—	11 per 10 HPFs	989
33 [10]	*	*	*	—	10 per 10 HPFs	518
34 [10]	*	*	*	—	12 per 10 HPFs	507
35 [12]	Malinois	9	M	—	4 per 10 HPFs	390

*: Individual data were not correlated to specific dogs; refer to the summary in the text. M = Male with unspecified castration status; MC = Castrated male; F = Female with unspecified ovariohysterectomy status; FS = Spayed female; NET G1 = Neuroendocrine tumor, grade 1; HPF = high power field; + = alive beyond the specified number of days. Dash (—) = Not reported.

**Table 2 vetsci-11-00371-t002:** Immunohistochemical profiles of canine GB NEN.

Dog	Chromogranin A+	NSE	Synaptophysin	Ki-67	CK7	Gastrin	S100	PGP 9.5
1 [18]	—	—	—	—	—	—	—	—
2 [39]	Y ^a^	—	—	—	—	—	—	—
3 [11]	3+	3+	—	—	—	—	—	—
4 [55]	Y	—	—	—	—	—	—	Y
5 [55]	Y	—	—	—	—	—	—	Y
6 [56]	3+	2+	1+	—	N	—	—	—
7 [57]	—	—	—	—	—	—	—	—
8 [58]	N	3+ ^a^	3+ ^a^	—	N	—	N	—
9 [58]	2+ ^a^	N	3+ ^a^	—	N	—	N	—
10 [59]	—	—	—	—	—	—	—	—
11 [62]	—	—	—	—	—	—	—	—
12 [44]	Y	Y	Y	0.04%	N	N	—	—
13 [44]	Y	Y	Y	0.05%	N	N	—	—
14 [60]	—	—	—	—	—	—	—	—
15 [60]	—	—	—	—	—	—	—	—
16 [60]	—	—	—	—	—	—	—	—
17 [60]	—	—	—	—	—	—	—	—
18 [60]	—	—	—	—	—	—	—	—
19 [60]	—	—	—	—	—	—	—	—
20 [60]	—	—	—	—	—	—	—	—
21 [60]	—	—	—	—	—	—	—	—
22 [10]	1+	2+	3+	—	—	2+	—	—
23 [10]	1+	1+	3+	—	—	3+	—	—
24 [10]	3+	3+	3+	—	—	3+	—	—
25 [10]	1+	1+	3+	—	—	3+	—	—
26 [10]	1+	3+	3+	—	—	3+	—	—
27 [10]	1+	3+	3+	—	—	3+	—	—
28 [10]	1+	2+	3+	—	—	3+	—	—
29 [10]	3+	3+	3+	—	—	3+	—	—
30 [10]	3+	3+	3+	—	—	3+	—	—
31 [10]	3+	3+	3+	—	—	3+	—	—
32 [10]	3+	3+	3+	—	—	3+	—	—
33 [10]	2+	3+	3+	—	—	3+	—	—
34 [10]	3+	-	3+	—	—	3+	—	—
35 [12]	—	3+ ^a^	—	—	—	—	—	—

Y = positive immunoreactivity with unspecified intensity; 1+ = positive immunoreactivity with minimal intensity; 2+ = positive immunoreactivity with moderate intensity; 3+ = positive immunoreactivity with strong intensity; N = negative immunoreactivity; Dash (—) = Not reported; ^a^ = intracytoplasmic immunoreactivity (unspecified for those that were not noted with a superscript).

## Data Availability

Not applicable.

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
