# Peer review of "Gallbladder Neuroendocrine Neoplasms in Dogs and Humans"

_vetsci, 2024, doi:10.3390/vetsci11080371_

Round 1

Reviewer 1 Report

Comments and Suggestions for Authors

Comments to Author:

 Overview

 The manuscript entitled “Gallbladder Neuroendocrine Neoplasms in Dogs and Humans” makes a very good revision of the histopathological and clinical aspects of neuroendocrine tumors in dogs in comparison with humans. The manuscript is very well written in an organized and coherent manner; the bibliography is updated and appropriate. There are a very few reports of these tumors in veterinary medicine and the authors made a collection of the available data in the literature and described the clinical, the histopathological characteristics and the median survival time of dogs with gallbladders neuroendocrine tumors, concluding that most of them have good behavior and long survival after surgery.

I only have a few minor comments:

Line 113: In the title, please add …. of neuroendocrine tumors.

Lines 137-140: please add references; describe the frequencies of each location; is there any risk factors described for both dogs and humans?

Line 152: the authors mentioned that cholelithiasis is frequently observed in humans with gallbladder NENs; is there any description regarding the percentage of cases that have history of cholelithiasis and these neoplasms?

Lines 185-188: please add references; please clarify if these markers are used for both species, humans and dogs.

Lines 189-201: this paragraph miss references

Line 211: insert the number of the reference instead (Rooper et al., 2017)

Lines 221-226: add literature references

Line 231: insert the number of the reference instead (Uccella et al., 2018)

Line 239: add literature references

Line 250: insert the number of the reference instead (Hirose et al., 2018).

Line 273: replace the word “where” by when available and insert (Table 1).

Author Response

Reviewer 1

Overview

 The manuscript entitled “Gallbladder Neuroendocrine Neoplasms in Dogs and Humans” makes a very good revision of the histopathological and clinical aspects of neuroendocrine tumors in dogs in comparison with humans. The manuscript is very well written in an organized and coherent manner; the bibliography is updated and appropriate. There are a very few reports of these tumors in veterinary medicine and the authors made a collection of the available data in the literature and described the clinical, the histopathological characteristics and the median survival time of dogs with gallbladders neuroendocrine tumors, concluding that most of them have good behavior and long survival after surgery.

I only have a few minor comments:

Response: Thank you for taking the time to review the manuscript and for providing helpful suggestions for its improvement.

Line 113: In the title, please add …. of neuroendocrine tumors.

Response: The text has been updated to “of neuroendocrine neoplasms”.

Lines 137-140: please add references; describe the frequencies of each location; is there any risk factors described for both dogs and humans?

Response: Reference citations have been included. The following has been added to the text “Canine thyroid tumors have a prevalence of 1-4%. In dogs, insulinomas comprise ~1-2% of all pancreatic neoplasms. Pheochromocytomas and paragangliomas account for 0.01 to 0.1% and 0.2%, respectively, of canine tumors. Pituitary tumors account for 13% of intracranial tumors in dogs. Less commonly, NENs have been documented in the hepatobiliary, gastrointestinal, and respiratory tracts. No definitive risk factors have been identified.”

Line 152: the authors mentioned that cholelithiasis is frequently observed in humans with gallbladder NENs; is there any description regarding the percentage of cases that have history of cholelithiasis and these neoplasms?

Response: The text has been edited to the following: “The fact that GB NENs are frequently observed in human patients with a history of cholelithiasis has been viewed as supporting evidence for the latter theory; while cholelithiasis is uncommon in dogs, 17.1% of the dogs described herein had choleliths concurrent with gallbladder NEN. Yet given the more recent revelations regarding the multipotent stem cell origin of most NENs, GB NENs may be derived from local progenitor cells rather than differentiated cells and O’Brien et al. and Morrell et al. have been cited.

Lines 185-188: please add references; please clarify if these markers are used for both species, humans and dogs.

Response: The following has been added to the text “…in humans and dogs”.

Lines 189-201: this paragraph miss references

Response: Reference citations have been added to the text.

Line 211: insert the number of the reference instead (Rooper et al., 2017)

Response: The error has been corrected.

Lines 221-226: add literature references

Response: Reference citations have been added to the text.

Line 231: insert the number of the reference instead (Uccella et al., 2018)

Response: The error has been corrected.

Line 239: add literature references

Response: Reference citations have been added to the text.

Line 250: insert the number of the reference instead (Hirose et al., 2018).

Response: The error has been corrected.

Line 273: replace the word “where” by when available and insert (Table 1).

Response: The text has been edited to “when available (Table 1)”.

Reviewer 2 Report

Comments and Suggestions for Authors

Richmond and colleagues

present reviews with comparative analysis of human and canine neuroendocrine tumors. The manuscript is well written and the literature in the dog on this topic is not much, so it is interesting for publication, but some things need to be fixed. Since there is a larger literature in humans than in dogs and much of this manuscript is on human tumors, it would be necessary to have a medical doctor among the authors so that he can accurately analyze the data present in the literature in humans.

Line 157: Cytological images of a NEN gallbladder would be desirable to depict the described features.

Paragraph 5.1: Are you talking about characteristics in the human species? This is unclear and should be specified.

Paragraph 5.2, Line 169: NEN tumors of the gallbladder are morphologically different in humans and dogs and these morphological differences should be well described. Human ones are described but not dog ones.

A histological image in both species would be desirable.

Paragraph 5.3: must always be specified when talking about markers used in humans.

Line 300-301: what is histological grading based on? What histological features are examined? This should be clarified.

Author Response

Reviewer 2

Comments and Suggestions for Authors

Richmond and colleagues present reviews with comparative analysis of human and canine neuroendocrine tumors. The manuscript is well written and the literature in the dog on this topic is not much, so it is interesting for publication, but some things need to be fixed. Since there is a larger literature in humans than in dogs and much of this manuscript is on human tumors, it would be necessary to have a medical doctor among the authors so that he can accurately analyze the data present in the literature in humans.

Response: Thank you for taking the time to review the manuscript and provide suggestions for its improvement. The authors respectively disagree with the reviewer’s final comment. The authors’ veterinary medical and biomedical educations enable accurate analysis of human medical literature.  

Line 157: Cytological images of a NEN gallbladder would be desirable to depict the described features.

Response: This is a review article, not a basic research publication. Never-the-less, the authors were unable to identify a reference with a cytologic image of a gallbladder NEN. Therefore, a cytologic image has been added to the manuscript (Figure 1).

Paragraph 5.1: Are you talking about characteristics in the human species? This is unclear and should be specified.

Response: The first sentence of section 5.1 has been revised to clarify that the characteristics apply to both species: “Neuroendocrine neoplasms across anatomic locations share similar cytologic features in humans and dogs.”

Paragraph 5.2, Line 169: NEN tumors of the gallbladder are morphologically different in humans and dogs and these morphological differences should be well described. Human ones are described but not dog ones.

Response: The text has been revised for clarity: “In humans and dogs, these neoplasms are composed of round to cuboidal cells arranged in one of several characteristic patterns, which include nests, rosettes, and solid cords interspersed by fibrovascular stroma. Well-differentiated NENs, which are more common in dogs, are composed of uniform polygonal cells, which can be arranged in solid nesting architecture, trabecular or ribbon-like morphology, or in glandular patterns”.

A histological image in both species would be desirable.

Response: This is a review article, not a basic research publication. Histologic images of gallbladder NEN, unlike cytologic images, are widely available in the human and veterinary literature. We have cited multiple manuscripts that include histologic images of gallbladder NENs.

Paragraph 5.3: must always be specified when talking about markers used in humans.

Response: The text of the first paragraph in section 5.3 has been revised and additional canine specific citations have been added: “A key characteristic enabling the identification of neuroendocrine cells is the presence of a distinct profile of cell markers. Broad-spectrum markers that can be used to identify neuroendocrine differentiation include chromogranin A (humans and dogs), synaptophysin (humans and dogs), and insulinoma-associated protein 1 (only validated in humans)”.

Line 300-301: what is histological grading based on? What histological features are examined? This should be clarified.

Response: The text has been edited and references added: “Histopathologic grading, based on the characteristics described in section 5.2 and the WHO guidelines, was reported for 3 patients (Table 1), including 1 described as a low-grade neuroendocrine carcinoma, and 2 NET-grade 1”.